# Molecular principles of assembly, activation, and inhibition in epithelial sodium channel

**Sigrid Noreng[1†], Richard Posert[1], Arpita Bharadwaj[2], Alexandra Houser[3], Isabelle Baconguis[2]***

[1]Department of Biochemistry and Molecular Biology, Oregon Health & Science University, Portland, United States; [2]Vollum Institute, Oregon Health & Science University, Portland, United States; [3]Neuroscience Graduate Program, Oregon Health & Science University, Portland, United States

**Abstract** The molecular bases of heteromeric assembly and link between Na$^+$ self-inhibition and protease-sensitivity in epithelial sodium channels (ENaCs) are not fully understood. Previously, we demonstrated that ENaC subunits – α, β, and γ – assemble in a counterclockwise configuration when viewed from outside the cell with the protease-sensitive GRIP domains in the periphery (Noreng et al., 2018). Here we describe the structure of ENaC resolved by cryo-electron microscopy at 3 Å. We find that a combination of precise domain arrangement and complementary hydrogen bonding network defines the subunit arrangement. Furthermore, we determined that the α subunit has a primary functional module consisting of the finger and GRIP domains. The module is bifurcated by the α2 helix dividing two distinct regulatory sites: Na$^+$ and the inhibitory peptide. Removal of the inhibitory peptide perturbs the Na$^+$ site via the α2 helix highlighting the critical role of the α2 helix in regulating ENaC function.

**\*For correspondence:**
bacongui@ohsu.edu

**Present address:** [†]Genentech, San Francisco, United States

**Competing interests:** The authors declare that no competing interests exist.

## Introduction

The ability to balance the amount of water inside and outside cells is absolutely essential for life. In the specialized epithelial tissues, the apical expression of the epithelial sodium channel (ENaC) gives rise to a transepithelial directional flow of Na$^+$ ions (*Palmer and Frindt, 1986*). ENaC function is therefore crucial in proper regulation of blood volume and pressure, as well as surface liquid volume in the respiratory and reproductive systems (*Boggula et al., 2018*; *Hummler et al., 1996*; *Rossier, 2014*). In humans, the essential role of ENaC in blood volume and pressure regulation is highlighted in gain-of-function mutations, as observed in Liddle syndrome, and also in loss-of-function mutations in pseudohypoaldosteronism type 1, severe genetic diseases that lead to hyper- and hypotension, respectively (*Palmer and Alpern, 1998*; *Liddle and Coppage, 1963*; *Shimkets et al., 1994*; *Cheek and Perry, 1958*; *Hanukoglu, 1991*; *Strautnieks et al., 1996*; *Chang et al., 1996*; *Edelheit et al., 2005*).

ENaC belongs to the ENaC/degenerin family, defined by Na$^+$-selectivity, voltage independence, and amiloride sensitivity (*Kellenberger and Schild, 2002*). Members of this family, including the well-studied relative Acid-Sensing Ion Channel (ASIC), have subunits that consist of short intracellular N- and C-termini, two membrane-spanning helices, and a large cysteine-rich extracellular domain (ECD) that can form homo- or heterotrimeric ion channels (*Jasti et al., 2007*; *Noreng et al., 2018*). In the case of ENaC, three homologous subunits, α, β, and γ, form trimers which are arranged in a counterclockwise direction when viewed from the extracellular space (*Noreng et al., 2018*; *Collier and Snyder, 2011*; *Collier et al., 2014*; *Chen et al., 2011*). Seminal cloning and functional studies of the ENaC subunits demonstrated that while homomeric α and diheteromeric forms of

ENaC containing α/β or α/γ can form functional ion channels, the α-β-γ presents robust $Na^+$ currents indicating that the triheteromeric form is the favored assembly (*Canessa et al., 1993*; *Canessa et al., 1994*; *Staruschenko et al., 2005*; *Lingueglia et al., 1993*; *Firsov et al., 1996*).

Unlike other ion channels, ENaC activity is primarily modulated by proteases that remove peptidyl tracts in the ECD (*Vallet et al., 1997*; *Sheng et al., 2006*; *Kashlan et al., 2015*). Removal of these polypeptides irreversibly converts ENaC channels from a low-channel-activity state to constitutively active channels (*Carattino et al., 2006*; *Bruns et al., 2007*). Canonically, the α subunit is cleaved twice by furin, while the γ subunit is cleaved once by furin and once by prostasin (*Carattino et al., 2006*; *Bruns et al., 2007*; *Hughey et al., 2004*; *Carattino et al., 2008a*; *Passero et al., 2010*). Of note, the β subunit does not have canonical protease sites. Conversely, extracellular $Na^+$ attenuates ENaC activity by binding to allosteric sites in the ECD, an effect referred to as $Na^+$ self-inhibition (*Fuchs et al., 1977*; *Awayda, 2016*). Interestingly, cleavage of the α subunit has been shown to abrogate $Na^+$ self-inhibition (*Sheng et al., 2006*). The molecular mechanisms of neither proteolytic activation nor $Na^+$ self-inhibition are currently understood.

We have previously solved the first structure of human ENaC at a nominal resolution of 3.9 Å by cryo-electron microscopy (cryo-EM) (*Noreng et al., 2018*). The structure provided valuable insight into channel assembly, stoichiometry and positions of the protease-sensitive domains, deemed the **G**ating **R**elease of **I**nhibition by **P**roteolysis (GRIP) domain. This initial study took advantage of ENaC constructs biochemically designed to be resistant to endogenous proteases, trapping the molecule in the uncleaved state. Our structure showed critical structural divergence from close relative ASIC in the peripheral region of the ENaC ECD, particularly in the finger and the specialized GRIP domains, which are not found in ASIC (*Jasti et al., 2007*). Here, we determined the structure of ENaC by single-particle cryo-EM at 3 Å to gain molecular insight into the roles of $Na^+$ and the protease-sensitive GRIP domains in ENaC function. The overall improvement of the map quality reveals for the first time the molecular source of the preferred channel assembly, and hints at mechanisms of $Na^+$ self-inhibition and proteolytic activation.

## Results

### Determinants of channel composition

To investigate the structural source of ENaC trimer assembly, we exploited a set of constructs, deemed ENaC$_{FL}$, which comprises wild-type α and β, and N-terminally eGFP-tagged γ, and behaves like wild-type ENaC as measured by electrophysiology (*Figure 1—figure supplement 1* and *Table 1*). We solved a 3 Å cryo-EM structure of ENaC$_{FL}$, based on the gold-standard Fourier shell correlation (*Figure 1—figure supplements 2–4*, *Table 2*). Resolution is higher in the channel core, calculated up to 2.6 Å, with β strands and smaller side chains clearly visible (*Figure 1—figure supplements 4* and *5*). To determine the structure of ENaC$_{FL}$, we expressed ENaC$_{FL}$ in HEK293T/17, solubilized in digitonin, and added two different Fabs, 7B1 (recognizes the α subunit) and 10D4 (recognizes the β subunit), to facilitate particle alignment (*Figure 1—figure supplement 1b,c*). Reference-free 2D class averages and 3D classifications reveal that ENaC$_{FL}$ channels form as α-β-γ

**Table 1.** IC$_{50}$ values of ENaC for three different blockers (amiloride, phenamil mesylate and benzamil).

IC$_{50}$ values (mean ± S.E.M) determined from dose-response curves for three different blockers (amiloride, phenamil mesylate and benzamil) at different holding voltages (-60 mV, -40 mV, -20 mV, 0 mV).

| | IC50 values (nM) | | |
| --- | --- | --- | --- |
| | Amiloride | Phenamil | Benzamil |
| 0 mV | 97.14 ± 21.62 | 51.37 ± 10.42 | 36.74 ± 13.25 |
| -20 mV | 80.05 ± 8.78 | 49.97 ± 11.18 | 29.41 ± 6.47 |
| -40 mV | 80.25 ± 11.37 | 43.37 ± 11.86 | 27.72 ± 6.65 |
| -60 mV | 86.34 ± 27.04 | 51.01 ± 14.12 | 32.90 ± 12.66 |

**Table 2.** Statistics of data collection, three-dimensional reconstruction, and model refinement.

| | | ENaC$_{FL}$ | |
|---|---|---|---|
| **Pre-merge dataset** | **1** | **2** | **3** |
| Material Source | Membrane | Whole cell | Whole cell |
| Detergent | Digitonin | Digitonin | Digitonin |
| Fab | 7B1 and 10D4 | 7B1 and 10D4 | 7B1 and 10D4 |
| Microscope | FEI Krios | FEI Krios | FEI Krios |
| Voltage (kV) | 300 | 300 | 300 |
| Detector | Gatan K2 Summit | Gatan K2 Summit | Gatan K2 Summit |
| Defocus range (µm) | −0.8 – −2.2 | −0.8 – −2.2 | −0.8 – −2.2 |
| Exposure time (s) | 3 | 3 | 3 |
| Dose rate (e−/Å$^2$/frame) | 1.0 | 1.0 | 1.0 |
| Frames per movie | 60 | 60 | 60 |
| Pixel size (Å) | 0.415 | 0.415 | 0.415 |
| Total dose (e−/Å$^2$) | 60 | 60 | 60 |
| Motion correction | UCSF MotionCor2 | UCSF MotionCor2 | UCSF MotionCor2 |
| CTF estimation | CTFFIND 4 | CTFFIND 4 | CTFFIND 4 |
| Particle picking | cryoSPARC blob | cryoSPARC blob | cryoSPARC blob |
| 2D/3D classification | cryoSPARC 2.11 | cryoSPARC 2.11 | cryoSPARC 2.11 |
| 3D classification and refinement | Relion 3.0, | Relion 3.0, | Relion 3.0, |
| | cryoSPARC 2.11, | cryoSPARC 2.11, | cryoSPARC 2.11, |
| | cisTEM 1.0 | cisTEM 1.0 | cisTEM 1.0 |
| Symmetry | C1 | C1 | C1 |
| Particles processed | 172 954 | 218 428 | 71 549 |
| Resolution masked (Å) | 3.57 | 3.05 | 3.96 |
| Map Sharpening B-factor (Å$^2$) | 91.8 | 87.3 | 97.9 |
| | | cryoSPARC 2.11 merged map | |
| Merged Symmetry | | C1 | |
| Merged particle count | | 252 071 | |
| Merged resolution masked (Å) | | 3.06 | |
| | | cisTEM 1.0.0 merged map | |
| Merged Symmetry | | C1 | |
| Merged particle count | | 248 079 | |
| Merged resolution masked (Å) | | 3.11 | |
| Initial model | | 6BQN | |
| Non-hydrogen atoms | | 11 740 | |
| Protein residues | | 1 594 | |
| Ligands (Na$^+$, NAG) | | 1, 10 | |
| Resolution (FSC = 0.143, Å) | | 3.06 | |
| Molprobity score | | 1.37 | |
| C$\beta$ deviations | | 0 | |
| Poor rotamers | | 0.84% | |
| Ramachandran outliers | | 0 | |

*Table 2 continued on next page*

*Table 2 continued*

| Pre-merge dataset | 1 | ENaC$_{FL}$ 2 | 3 |
|---|---|---|---|
| Ramachandran allowed | | 2.7% | |
| Ramachandran favored | | 97.3% | |
| Bond length rmsd (Å) | | 0.002 | |
| Bond angle rmsd (°) | | 0.390 | |

counterclockwise when viewed from outside the cell (*Figure 1—figure supplements 2–4*). However, the transmembrane domain (TMD) and the cytosolic domain (CD) were not resolved; we speculate that preferred particle orientation, air-water interface denaturation, and intrinsic protein flexibility and conformational heterogeneity contribute to the lack of 3D reconstruction of the TMD and CD (*Figure 1—figure supplement 4f*). Therefore, we did not include the TMD and CD portions in the ENaC$_{FL}$ structure (*Figure 1—figure supplement 4*). The higher resolution of ENaC$_{FL}$ structure affords us confidence in the placement of side chains for the first time, providing unprecedented insight into how the ECD mediates ENaC function.

It is known that functional ENaC channels require at least one α subunit (*Canessa et al., 1994*; *Fyfe and Canessa, 1998*; *McNicholas and Canessa, 1997*). Additionally, because the γ subunit gene contained the purification tag, all purified ENaCs contain at least one γ subunit (*Figure 1—figure supplement 1a*). Thus, if other combinations of ENaC heteromers were present, classes with one (α-γ-γ) or two Fabs (α-γ-β or α-α-γ) forming a 35° and 120° angle about the pseudo three-fold axis, respectively, would be observed (*Figure 1a*; *Stewart et al., 2011*; *Baldin et al., 2020*). However, no such classes were detected (*Figure 1—figure supplements 2*, *3* and *4a*). To understand how ENaC favorably assembles as a heterotrimer with α-β-γ arranged counterclockwise, we inspected molecular interactions in the ECD at the subunit interface formed by the finger (α1 and α2 helices in all three subunits), the knuckle (α6 helix in all three subunits), and the GRIP domain (*Figure 1*). All subunit interfaces share van der Waals interactions between the first two helical turns of the α2 helix and the α6 helix of the adjacent subunit. Additionally, these α2 helices are capped by conserved serine residues (*Figure 1—figure supplement 5*).

By contrast, the interfaces formed by the α1 helix of one subunit and the α6 helix of the adjacent subunit show notable differences in both nonpolar and polar interactions. First, nonpolar contacts involve a tyrosine only found in α and γ; the equivalent residue is a leucine in β (βLeu127). The αTyr162 is surrounded by the hydrophobic αLeu161 and βVal474 (*Figure 1c*). The equivalent γTyr129, however, is tucked further into its own subunit, in a pocket comprising residues from the γ-α1 helix, γ-α2 helix, and γGRIP domain, as well as the adjacent αMet505 (*Figure 1e*). The nonpolar interactions at β/γ interface present yet another combination, in which two hydrophobic residues, βIle126 and βLeu127, make multiple hydrophobic contacts with the γ-α6 helix. In a conformation distinct to this interface, γTrp486 is wedged between the C-terminal end of the β-α1 helix and the βGRIP domain loop, locking the residue in place (*Figure 1d*). This conformation would result in a clash if the β and γ subunits were swapped, indicating that the positions of the aromatic residues may play a large role in defining the counterclockwise arrangement of channel subunits.

Second, polar interactions via hydrogen bonds are only found at two interfaces. The α/β interface αTyr162 is also poised to participate in a hydrogen bonding network with neighboring αArg190 in the αGRIP domain and βGlu478 (*Figure 1c*). Thus, the α subunit acts as a hydrogen bond donor to the β subunit (*Figure 1f*). However, at the γ/α interface, the γ subunit is a hydrogen bond acceptor, with the backbone carbonyl oxygens of γGly130 and γPhe131 forming hydrogen bonds with the guanidino group of αArg508 (*Figure 1e*). Finally, there is no clear hydrogen bond network at the β/γ subunit interface. Thus, the hydrogen bond networks at the different interfaces confer specificity for the counterclockwise α-β-γ channel (*Figure 1f*).

We extended our analysis of homomeric channels by generating in silico models of homomeric forms of each ENaC subunit. To generate homomeric α, β, and γ channels (α$_{homo}$, β$_{homo}$, and γ$_{homo}$), we used the coordinates of the ENaC$_{FL}$ structure, assuming C3 symmetry around the three-fold axis (e.g. α$_{homo}$, *Figure 2a*). We believe that this is a reasonable assumption, based on structures of the

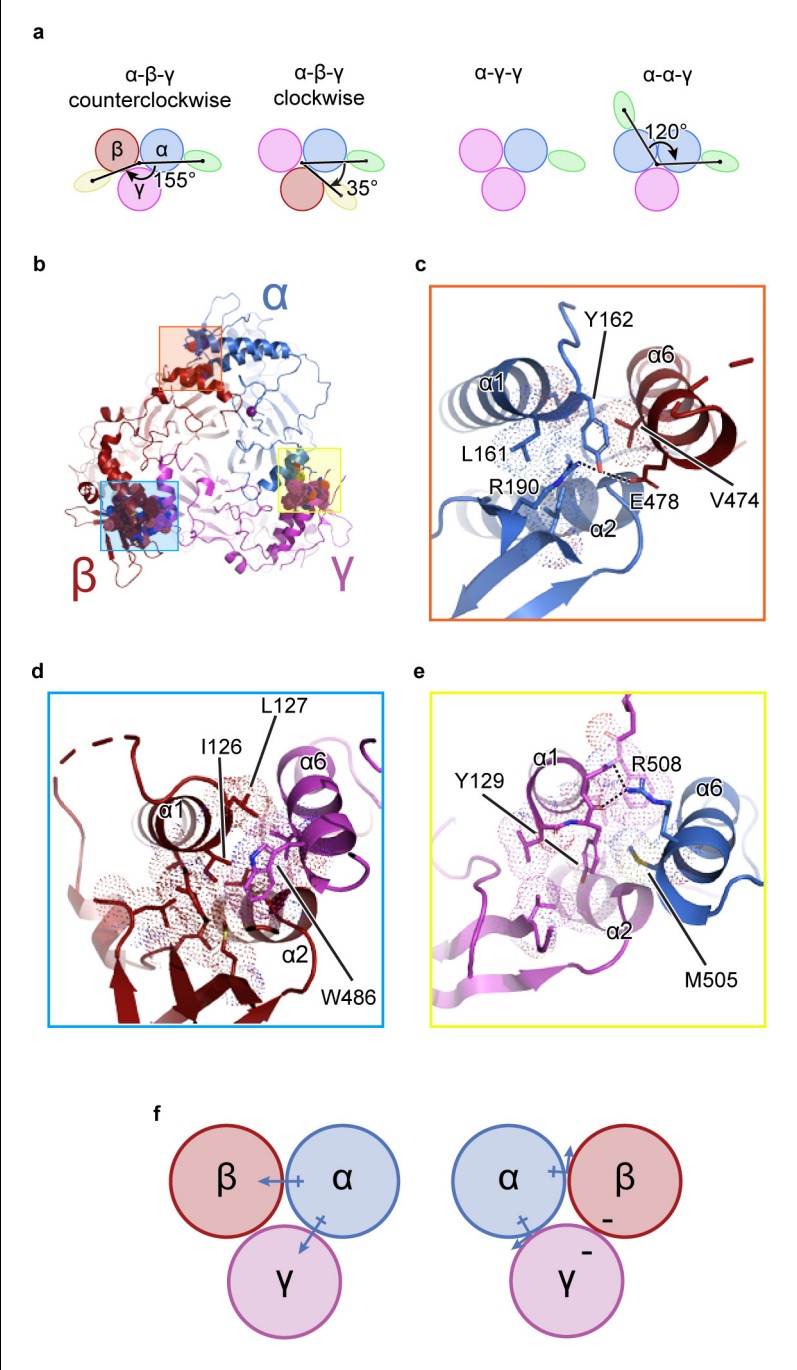

**Figure 1.** The unique molecular interactions at the subunit interface define heteromeric assembly of ENaC.
(a) Top-down cartoon schematic illustration of ENaC with α-β-γ counterclockwise as resolved by cryo-EM (top left) and three possible assemblies of ENaC based on the defined purification scheme (see Materials and methods) as seen from left: α-β-γ clockwise (second panel), α-γ-γ (third panel), and α-α-γ (fourth panel). Subunits and Fabs are colored blue (α), red (β), magenta (γ), green (7B1) and yellow (10D4). (b) View of the ENaC_{FL} from the extracellular side and shown in cartoon representation. The α, β, and γ are colored blue, red, magenta, respectively. Boxed regions define subunit interactions near the top of the ECD. (c) Close-up view of the α-β interface as highlighted with an orange square in (b). The hydroxyl group of αTyr162 forms hydrogen bonds with αArg190 and βGlu478. Dashed lines indicate distances of 2.5–3.5 Å. (d) Zoomed-in view of the β-γ interface in blue boxed region. The equivalent residue βLeu127 is primarily interacting with residues in the adjacent α6. Instead, βIle126 resides in the equivalent position as in αTyr162 and γTyr129 makes van der Waals contacts with the residues from the α2, βGRIP, and the adjacent α6. (e) Enlarged view of the γ-α interface, yellow boxed region. The side chain of the equivalent

*Figure 1 continued on next page*

*Figure 1 continued*

γTyr129 is largely surrounded by hydrophobic residues. (f) Cartoon schematic illustration of the ENaC hydrogen bonding network. The α subunit donates hydrogen bonds to both the β and γ subunits in the counterclockwise arrangement (left). If the positions of α and β are swapped, the hydrogen bond donors and acceptors are mutually inaccessible (right).

The online version of this article includes the following figure supplement(s) for figure 1:

**Figure supplement 1.** Biochemical and functional characterization of ENaC$_{FL}$.

**Figure supplement 2.** Cryo-EM initial data processing workflow.

**Figure supplement 3.** Cryo-EM data processing for the final map.

**Figure supplement 4.** Cryo-EM analysis of ENaC$_{FL}$ dataset.

**Figure supplement 5.** Cryo-EM potential maps of different regions in ENaC$_{FL}$ map.

**Figure supplement 6.** Stereoview of cryo-EM potential maps of the GRIP domain in ENaC$_{FL}$ map.

closely related ASIC. Comparison of these homomeric models reveals steric clashes in both the distal (finger/knuckle domain interface, *Figure 2b–d*) and core (lower palm/thumb domain interface, *Figure 2e–g*) ECD of the β and γ subunits. Focusing on the distal ECD, the β$_{homo}$ channel α6 and α2 helices are 3 Å closer than the α$_{homo}$ channel (*Figure 2b*), pointing to potential steric clash in the interface. The γ$_{homo}$ channel appears even less stable in this region, with α6 and α2 clearly intersecting (*Figure 2c*). Conversely, in the core ECD, the β$_{homo}$ channel shows clear steric clash between the β11-β12 linker and the adjacent β10 strand (*Figure 2e*). The core ECD of the γ- and α subunits are similar (*Figure 2f*) and, interestingly, the β11-β12 linkers are both similar to that of the ASIC open and closed states (*Baconguis and Gouaux, 2012*; *Baconguis et al., 2014*; *Yoder et al., 2018*). Meanwhile, the β subunit β11-β12 linker resembles that of ASIC trapped in the desensitized state (*Gonzales et al., 2009*). However, the functional consequences of the β11-β12 linker asymmetry, when comparing all three subunits, have not been investigated in detail, so caution is required in interpreting the state of the β11-β12 linker in the β subunit. None of these steric clashes are obvious in the α$_{homo}$ channel, as expected given this channel's ability to pass current in vitro.

We next determined how the domains within β and γ subunits arrange to give rise to steric clashes. To this end, we performed an alignment of the structure of each subunit by their highly similar upper palm domain. This alignment revealed a rigid-body shift of the finger (α1 and α2 helices) and thumb (α4 and α5 helices) domains in both β and γ subunits relative to the α subunit (*Figure 2—figure supplement 1a–c*). To determine the consequences of the shift in β and γ, we measured the distances between the Cα atoms of the conserved tryptophan residue in finger domain α2 helix in the homomeric models. The region is suitable for this analysis due to its greatly increased local resolution compared to the overall structure (*Figure 1—figure supplements 4d* and *5*). Compared to ENaC$_{FL}$, the distances between the Cα atoms of homomeric models, especially γ$_{homo}$, are shorter (*Figure 2—figure supplement 1d–g*). As a result, the subunits are compressed toward the three-fold axis, causing major steric clashes.

## Identification of a putative Na$^+$ binding site

We observed a map feature located near the β-ball domain and the β6-β7 loop of the α subunit, where residues αGlu335, αAsp338, and αSer344 in the α-β6-β7 loop have been identified as important in Na$^+$ self-inhibition (*Figure 3a,b*; *Kashlan et al., 2015*). The map quality in this region is estimated to be well beyond 3 Å and thus the positions of the side chains are reliable. We speculate that this map feature is a cation, perhaps a Na$^+$ ion, based on the surrounding residues, the above-mentioned functional studies, and the presence of high Na$^+$ (150 mM) during purification, (*Figure 3b*). The cation interacts closely with several negative charges: the carboxyl group of αAsp338, and the carbonyl oxygens of αLeu135, αGlu335, and αVal346; all of these interactions are within distances of 2.5 – 3.5 Å. The hydroxyl group of αSer344 likely interacts with the cation via a water molecule, at a distance of approximately 4 Å. While these measured distances suggest that this feature is a positively charged ion, the cation site is perhaps not very highly selective for Na$^+$. This is consistent with the ability of other cations like K$^+$ and Li$^+$ to reduce ENaC macroscopic currents, although the inhibitory effect of Na$^+$ is larger in comparison (*Kashlan et al., 2015*; *Bize and Horisberger, 2007*). Indeed, definitive identification of this feature as the Na$^+$ self-inhibition site

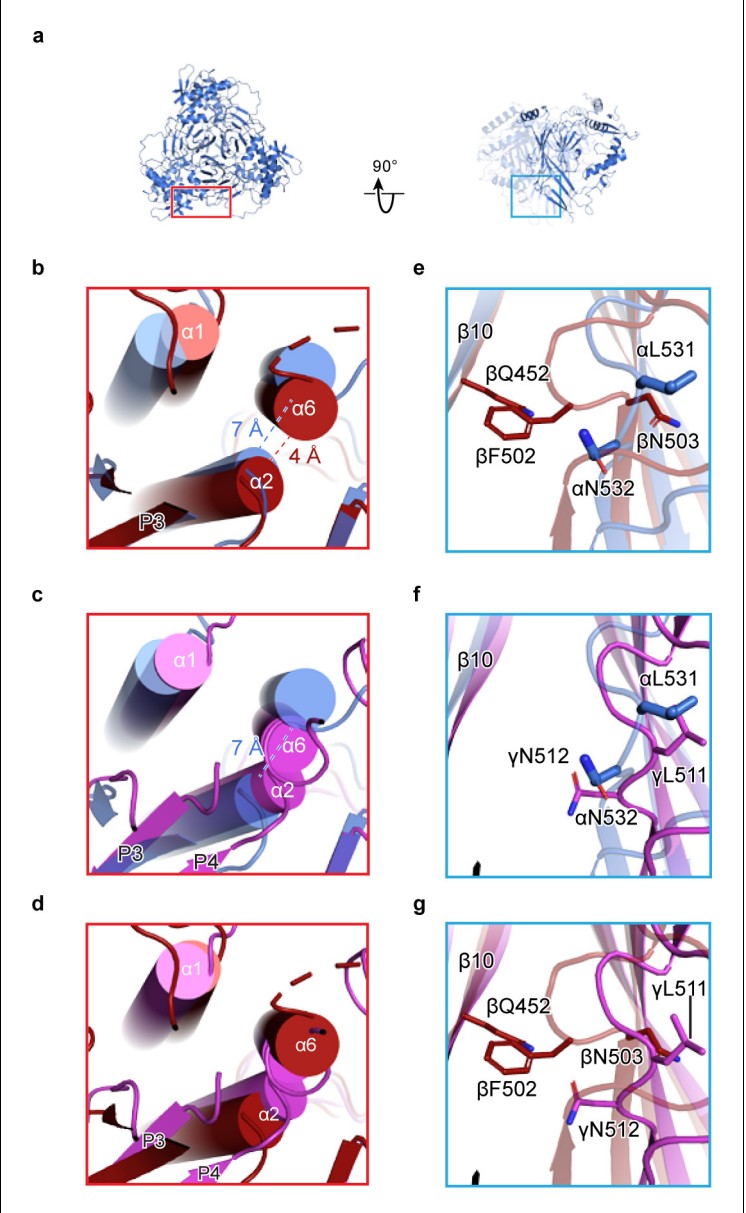

**Figure 2.** Human ENaC is a heteromeric channel with three different subunits. (a) Generated model of homomeric α ENaC using coordinates of the α subunit from the ENaC$_{FL}$ structure. The two additional α subunits that complete the trimer were generated around the three-fold axis of symmetry. (b–d) Enlarged view of the subunit interface, from the red rectangular frame in (a), focusing on the α1 and α2 helices of the finger domain from one subunit and the α6 helix of the knuckle domain from the adjacent subunit. The homomeric trimers of α, β, and γ are superposed using the coordinates of the upper palm domain. Cartoon cylinders are colored as in (*Figure 1*). The α2 and α6 helices are spatially accommodated in the homomeric α (b and c) while minor and major clashes are observed in homomeric β (b and d) and γ (c and d), respectively. (e–g) Close-up view of the β10 strand from one subunit and the β11-β12 linker from the adjacent subunit. The observed conformation of the α (e and f) and γ (f and g) linkers is reminiscent of the β11-β12 linker conformations of the open and closed states of chicken acid-sensing ion channel 1. Conserved leucine and asparagine residues comprise the β11-β12 linker. The adopted linker conformation in β (e and g) is similar to that of the desensitized state of cASIC1. In this conformation, there is a steric clash between Gln452 of β10 and Phe502 of the β11-β12 linker.

The online version of this article includes the following figure supplement(s) for figure 2:

**Figure supplement 1.** The domains within the ENaC subunits favor a heteromeric assembly.

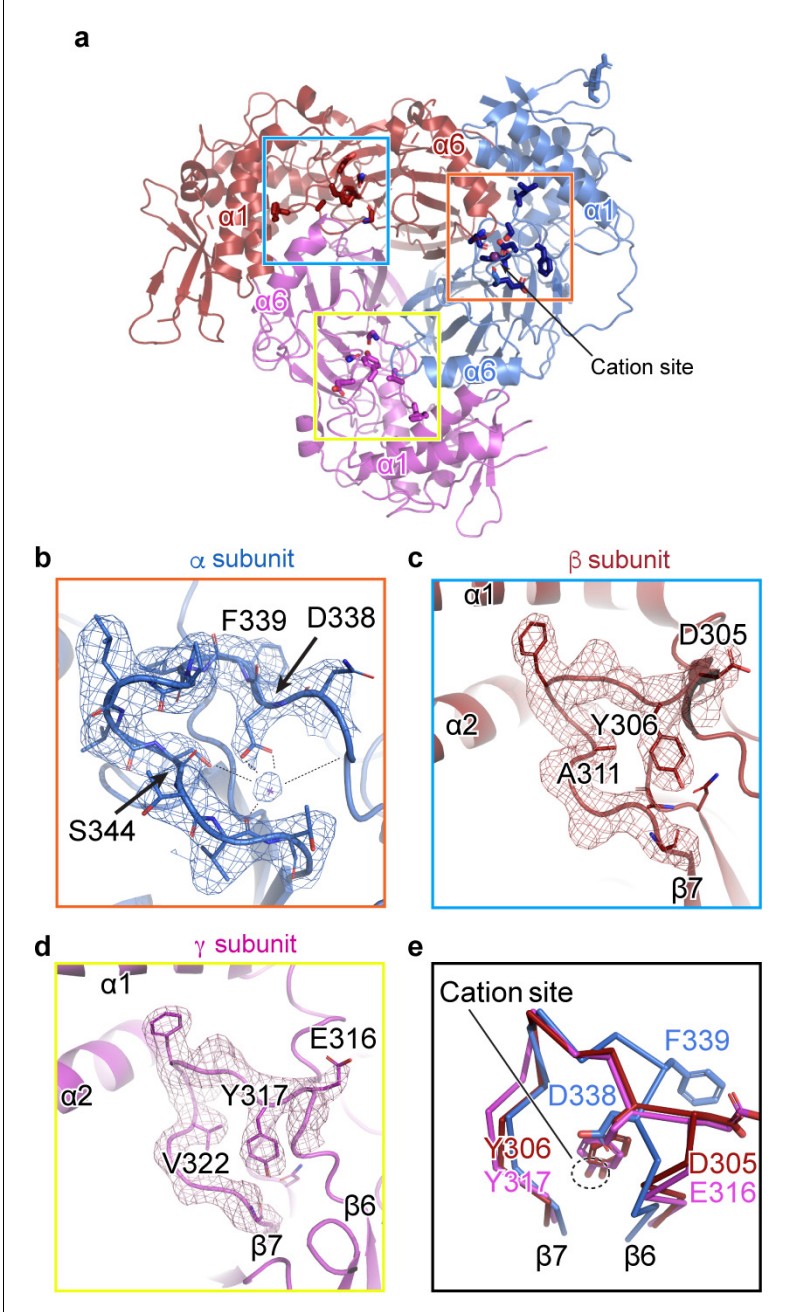

**Figure 3.** A cation binding site is located in the β6-β7 loop of the α subunit finger domain. (**a**) Cartoon representation of ENaC perpendicular to the membrane. α, β and γ are colored blue, red and magenta, respectively. The orange box shows the region of the cation site that is speculated to be a Na$^+$ ion in the α subunit (**b**). The blue and yellow box represent the equivalent region, not occupied by a cation, in β (**c**) and γ (**d**) subunit, respectively. (**b–c**) Model of the β6-β7 loop superimposed with the potential map for the α subunit (**b**), β subunit (**c**) and γ subunit (**d**). (**b**) View of the cation site in the α subunit. The electrostatic potential of the β6-β7 loop and the Na$^+$ is shown as a mesh. Dashed lines indicate distances of 2.5 – 3.4 Å. Residues shown with dashed lines are Leu135, Glu335, Asp338, and Val346. Ser344 is 3.8 Å away from the Na$^+$. (**c, d**) Views of the equivalent regions in β (**c**) and γ (**d**). The residues occupying the equivalent position as Ser344 in α are alanine in β and valine in γ. (**e**) Superposition of the β6-β7 loops of all subunits demonstrate a striking difference in conformation in the α subunit. The acidic αAsp338 faces towards the Na$^+$ site and Phe339 faces away from the cation site. The residues in the β and γ loops adopt a swapped conformation relative to the α subunit in which the aromatic residues are facing the equivalent sites while the acidic residues are exposed in solution.

would require resolving the structure of ENaC in the presence of $K^+$ and determining if there are any associated structural changes.

We next examined the related positions in the β and γ subunits for a similar feature. The pocket into which $Na^+$ binds in the α subunit is instead occupied by the side chains of βTyr306 and γTyr317 in their respective subunits (*Figure 3c,d*). Moreover, where αSer344 contributes a favorable polar interaction to the binding site, the equivalent positions in the β- and γ subunits are aliphatic (βAla311 and γVal322). In all three subunits, there is an acidic-aromatic residue pair at the segment of the β6-β7 loop believed to modulate $Na^+$ self-inhibition. Superposition of this loop from all three subunits reveals that the α subunit adopts a swapped conformation relative to the β- and γ subunits near the putative cation binding site (*Figure 3e*). The acidic residues in the β- and γ subunits are exposed to solution, while the tyrosine hydroxyl groups are buried, participating in a network of internal hydrogen bonds. A phenylalanine in the equivalent position of the α subunit is incapable of participating in these hydrogen bonds and may explain the different conformation of the loop and, thus, the existence of the cation binding site.

## Characterization of the inhibitory peptide binding sites and GRIP domains

Proteolytic processing is one of the defining characteristics of ENaC gating, in which the removal of inhibitory P1 peptides, located in the α and γ subunit, shifts ENaC from a low to a high $P_o$ state (*Carattino et al., 2008a*; *Passero et al., 2010*; *Figure 4*). In the α subunit, the P1 peptide consists of residues α184–191 (LPHPLQRL) while the γ subunit P1 peptide includes γ153–163) RFSHRIPLLIF (*Figure 4a,f*). Because the residue numbers of the inhibitory P1 peptides vary across different species, we propose a numbering system in which the highly-conserved prolines (αPro187, βPro149, and γPro159) are denoted as position *0* to simplify discussion. Residues closer to the N-terminus are labeled as *-n*, while residues closer to the C-terminus are labeled as *+n*, for example αGln189, βVal151, and γLeu161 are each *+2* (*Figure 4b,d*). This naming convention echoes that of protease binding sites (*Schechter and Berger, 1968*). The N-termini of the P1 peptide are highly diverse in primary sequence, molecular organization, and interaction. The αP1 contains a histidine at *−1*, which inserts into a pocket formed by the thumb, finger, and P3 strand of the α subunit. A proline at *−2* changes the direction of the αP1 peptide, pointing the *−3* leucine toward the top of the α1 helix, which anchors the αP1 peptide between α1 and α2 helices (*Figure 4b*). The βP1 ($N_{-3}H_{-2}T_{-1}$), on the other hand, forms a short, helical structure that is stabilized by a network of aromatic residues from both the α1 helix and the βGRIP domain (*Figure 1—figure supplement 6*). Finally, the γP1 ($R_{-6}F_{-5}S_{-4}H_{-3}R_{-2}I_{-1}$) binds a hydrophobic pocket in the thumb, finger, and P3 strand with its *−1* residue, just as in αP1. However, unlike αP1, γP1 has a solvent-exposed arginine at *−2* instead of a proline (*Figure 4d* and *Figure 1—figure supplement 6e*). Thus, the γP1 does not have the conformational constraint present in the αP1 that is introduced by a proline. Instead, we observe a clear map feature of γP1 that is extended alongside the finger domain in which the main chain and residues in γP1 forge multiple interactions with the thumb domain of the γ subunit (*Noreng et al., 2018*).

The C-terminal side of the P1 peptide primarily makes contact with the finger domain and the bulk of the GRIP domain. There is sequence divergence in the α subunit, with $Q_{+2}R_{+3}L_{+4}$ as opposed to the hydrophobic sequences in the β ($V_{+2}L_{+3}I_{+4}$) and γ ($L_{+2}I_{+3}F_{+4}$) subunits. Additionally, each P1 peptide contains a conserved leucine residue at *+1* which forms hydrophobic contacts with a highly-conserved tryptophan from the α2 helix of the finger domain in all three subunits (αTrp251, βTrp218, and γTrp229, *Figure 4c,e*).

## Investigation of the GRIP domains

The first structure of ENaC, referred to as ΔENaC which comprised of subunits with truncated amino and carboxy termini and other mutations in the ECD, demonstrated that all GRIP domains, including the protease-insensitive βGRIP, adopt similar anti-parallel β strand architecture (*Noreng et al., 2018*). The P3 and P4 strands of the GRIP domain (especially αGly225 and αThr240) have an outsize role in reduction of mouse ENaC current upon binding of the inhibitory peptide (*Kashlan et al., 2010*). In our ENaC$_{FL}$ structure, the P3 and P4 strands are linked by a loop containing a predicted glycosylation site adjacent to the α5 helix of the thumb domain in all three subunits. Additionally,

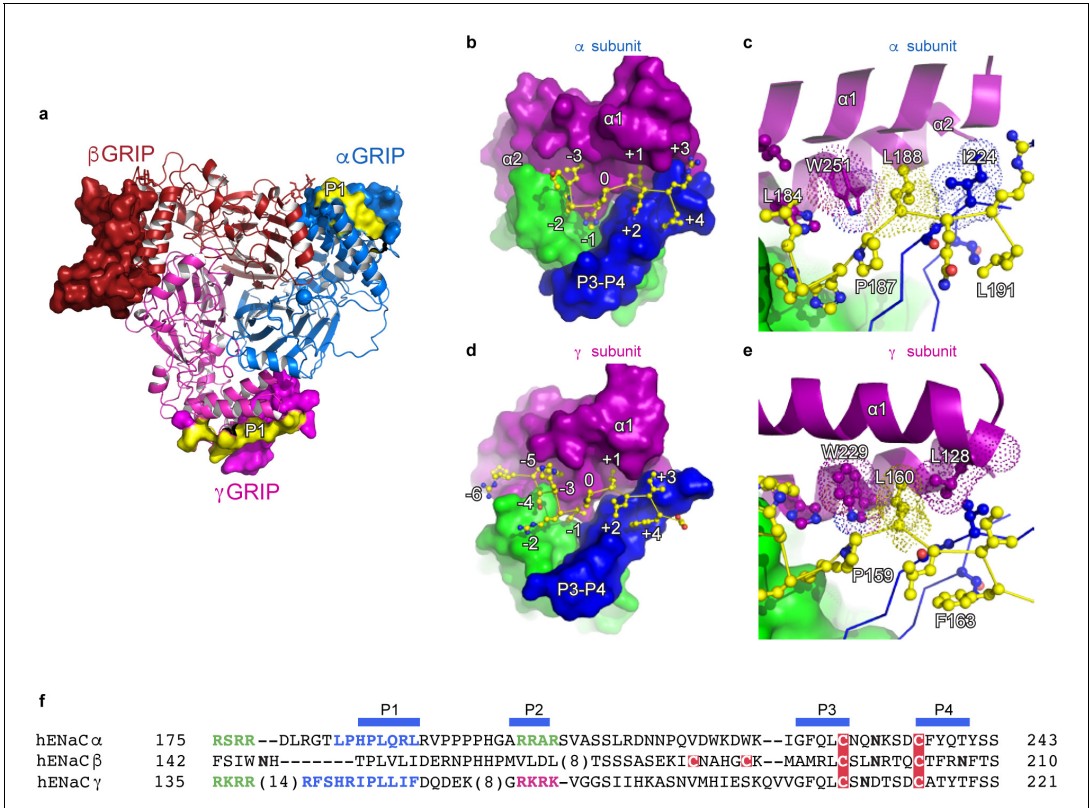

**Figure 4.** The inhibitory peptides in α and γ interact distinctly with the gating domains. (**a**) Cartoon representation of ENaC perpendicular to the membrane where the GRIP domains in all three subunits are shown as surface representation. α, β and γ are colored blue, red and magenta, respectively. Additionally, the inhibitory P1 peptides of αGRIP and γGRIP are highlighted in yellow. (**b**) Overall view of the inhibitory peptide pocket in the α subunit. The finger, thumb, and P3-P4 strands of the GRIP domains are colored purple, green, and blue, respectively, and shown in surface. The inhibitory peptide is shown in sticks representation and colored yellow. In the α subunit, the inhibitory peptide only extends to the −3 position. (**c**) Close-up view of the pocket consisting of conserved residues Pro187, Leu188, and Trp251. Leu188 makes hydrophobic contacts with Trp251 from the α2 helix and Ile224 of the P3 strand of the GRIP domain. (**d**) Overall view of the inhibitory peptide pocket in the γ subunit. Representations are similar to (**b**). In the γ subunit, the inhibitory peptide extends to position −6 making more extensive contacts with the finger and thumb domains. (**e**) Close-up view of the binding pocket consisting of the equivalent residues shown in (**c**). In the γ subunit, the Leu160 primarily interacts with the residues in the α2 helix. The residues in the GRIP domain that interact with the inhibitory peptides are in sticks representation in panels (**c**) and (**e**). (**f**) Sequence alignment of ENaC GRIP domains (hENaCα, GenBank ID:4506815; hENaCβ, 124301096; hENaCγ, 42476333). The sequences were aligned with Clustal Omega and manually adjusted. Coloring or shading is as follows: cysteines participating in disulfide bonds are in red boxes, glycosylation sites (predicted and/or present in cryo-EM map) are shown as bold N, furin sites are in green, prostasin site in magenta, and the P1 peptides in α and γ are in blue.

the important residue αGly225 is adjacent to αThr240 and forms hydrogen bonds with the C-terminal end of the αP1 peptide (*Figure 5b*; *Blobner et al., 2018*).

To further investigate the α and γ GRIP domains, we assayed the cleavage state of our sample by SDS-PAGE and western blot (*Figure 1—figure supplement 1c–f*). Our sample had partially proteolyzed α and γ subunits, as expected given the wild type GRIP domain. Using focused classification, we aimed to identify the different cleavage states - absence or presence of the inhibitory peptide – that our SDS-PAGE analysis suggests to exist. Particles from the final 3D refinement in both cryoSPARC and cisTEM were subjected to focused classification in cisTEM, centered on the αGRIP domain (*Figure 5—figure supplement 1*; *Punjani et al., 2017*; *Grant et al., 2018*). Assuming that the αGRIP domain is only cleaved at the canonical protease sites, there are four major combinatorial classes: uncleaved, fully cleaved, and two cleavage states at either protease sites. We requested five classes in each focused classification to allow for some heterogeneity in the particles.

In the αGRIP classification, the largest class (50% of the total number of particles) demonstrated features similar to ΔENaC, which could not be cleaved. We thus consider this the uncleaved class (*Figure 5a,b* and *Figure 5—figure supplement 1d*). We merged two classes which lacked features

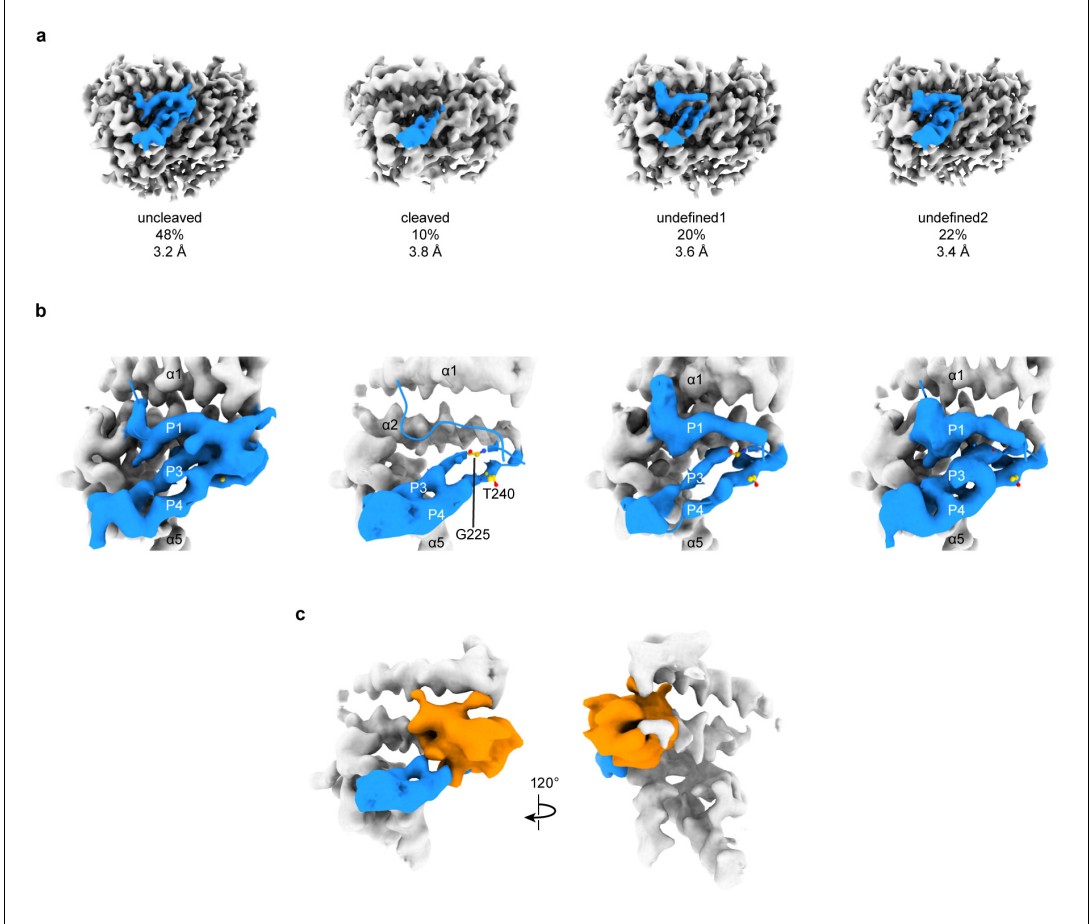

**Figure 5.** 3D focused classification of the inhibitory peptide pocket in the α subunit reveals important site for ENaC regulation. (**a**) 3D classification of the GRIP domain in the α subunit revealed four major classes. Two classes clearly demonstrate the uncleaved and cleaved states of the α subunit representing 48% and 10% of the particles. Five classes were initially requested. The αGRIP domain is colored in blue. The remaining region of the ENaC map is colored gray for simplicity. (**b**) Close-up view of the GRIP domain. Compared to the uncleaved state, the map of the cleaved state shows that the region near the P3-P4/α2 is more disordered based on the lack of well-defined features that are observed in the uncleaved state. Residues that have been identified to markedly reduce peptide inhibition when mutated to tryptophans, Gly225 and Thr240, are shown in yellow and represented in sticks. (**c**) The difference map (orange) overlaps with the region in the GRIP domain that is absent or more disordered in the cleaved state.

The online version of this article includes the following figure supplement(s) for figure 5:

**Figure supplement 1.** Cryo-EM data analysis of all α-GRIP classes in cryoSPARC.

**Figure supplement 2.** Cryo-EM data analysis of all γ-GRIP classes in cryoSPARC.

**Figure supplement 3.** 3D focused classification of the inhibitory peptide pocket in the γ subunit demonstrates heterogeneity in the inhibitory peptide cleavage states.

of the inhibitory peptide and the P3 strand into the fully-cleaved class (*Figure 5a,b* and *Figure 5—figure supplement 1b*). The fully-cleaved class contains 10% of the total particles. The other two classes, comprising 40% of the total particles, had a similar overall potential map to the uncleaved class, but contained some differences along the inhibitory peptide and P3 strand. We consider these classes undefined, and believe that they likely are an intermediate cleavage state or too low-contrast for proper classification (*Figure 5a,b* and *Figure 5—figure supplement 1c,e*). While on the one hand the western blot analysis showed a large population of cleaved α subunit, on the other hand, the focused classification analysis demonstrated a small population of the fully cleaved class. We speculate that there are three major reasons for the discrepancy. First, the existence of the relatively large class of undefined molecules in which the cleavage state of the P1 peptide is unresolved could contribute to the discrepancy between the observed intensity of the cleaved α band observed in

western blot analysis (*Figure 1—figure supplement 1d*). Second, due to the binding site of our antibody spanning both cleavage sites, what appears to be a single large band may in fact represent both partially-cleaved and fully-cleaved α subunit (*Noreng et al., 2018*). And third, the set of particles used for focused classification was derived from rounds of 2D and 3D classifications, which removed denatured complexes and particles in thick ice, as examples. Thus, the population of particles used for SDS-PAGE and western blot analyses is not the same as the population used for focused classification. Nevertheless, implementing focused classification resulted in 3D maps that demonstrate differences in map features in the GRIP domain. A difference map between the cleaved and uncleaved maps shows a prominent feature overlapping the position of the uncleaved inhibitory strand, as expected (*Figure 5c*). The difference map also highlights deviations in the P3 strand potential, in agreement with observations in the cleaved and undefined classes indicating increased flexibility of this region upon cleavage. This disordered region begins near αGly225 in αP3 (*Figure 5b*). We thus speculate that αP3 becomes more mobile when the inhibitory peptide is proteolytically removed.

The γ subunit is known to be cleaved by several proteases aside from the canonical furin and prostasin, the latter of which is not present in our expression system (*Kleyman et al., 2009*). We expect these non-canonical cleavages, if present, to segregate into the undefined class. All five classes still showed features of the γ-inhibitory peptide (*Figure 5—figure supplements 2* and *3*). There are detectable differences in the inhibitory peptide between the five classes, with class one showing the most striking difference from the overall structure (*Figure 5—figure supplement 3*). We observed a small reduction of electron potential at the C-terminus of the inhibitory peptide and the γP3 strand, as expected. Nevertheless, this analysis suggests that the vast majority of the particles used for the initial map contain intact γGRIP domains.

## 7B1 Fab binds to the uncleaved and cleaved states of αGRIP

Given that all of the classes (cleaved, uncleaved, and undefined of both α and γ subunits) have the same overall topology, we more closely investigated Fab binding and its effect on ENaC activity. 7B1 binds primarily to the finger domain and finger/thumb interface of the α subunit (*Figure 6*). 7B1 map feature at the αECD is equally strong in both the cleaved and uncleaved states of αGRIP and the cleaved and partially-cleaved states of γGRIP (*Figure 5—figure supplements 1* and *2*). Additionally, we did not observe any structural rearrangements in between the two states (*Figure 5c*), which does not align with the proposed gating mechanism derived from the structures of different functional states of ASIC (*Yoder et al., 2018*). It is possible that 7B1 traps ENaC in the conformation natively adopted by the uncleaved channel, regardless of the actual state of the channel. We thus assayed ENaC current before and after application of 100 nM 7B1 (10-fold greater than the observed $K_D$ for purified $ENaC_{FL}$, data not shown). If 7B1 traps the ECD in the uncleaved state, channel current would be low after application of protease in the presence of 7B1. We did not detect measurable acute differences in current magnitude or profile upon addition of 7B1 to either closed or open channels (*Figure 7a,b*). We confirmed surface binding by confocal microscopy of cells expressing $ENaC_{FL}$ (*Figure 7c*). We can thus conclude that 7B1 binds ENaC at the cell surface, and that this binding does not reduce or modulate the macroscopic ENaC currents.

We also tested whether 7B1 binds only uncleaved ENaC. We performed fluorescence-detection size-exclusion chromatography (FSEC) and western blots of purified $ENaC_{FL}$, either as-purified (mostly uncleaved) or treated with trypsin. Additionally, to assay whether 7B1 can bind any ENaC or only $Na^+$-bound ENaC, we performed these experiments with an additional variable of $Na^+$ or $K^+$ buffer, the latter of which should not induce a $Na^+$-bound conformation (*Figure 7—figure supplement 1*). As expected, uncleaved $ENaC_{FL}$ binds both 7B1 and 10D4 (*Figure 7—figure supplement 1*). These results are in agreement with prior studies on ΔENaC. Purifying and analyzing the protein in $K^+$ buffer showed no apparent difference in binding behavior. We thus conclude that 7B1 is able to bind uncleaved $ENaC_{FL}$ in both high- and low-$Na^+$ conditions. Encouragingly, this trend holds after $ENaC_{FL}$ is treated with trypsin, moving the channels into a cleaved state (*Figure 7—figure supplement 1*). In summary, 7B1 can bind ENaC regardless of cleavage or $Na^+$ concentration and does not modulate ENaC current. Thus, we believe that 7B1 does not trap ENaC in a closed-like conformation, and our classifications of the cleaved αGRIP structures are valid.

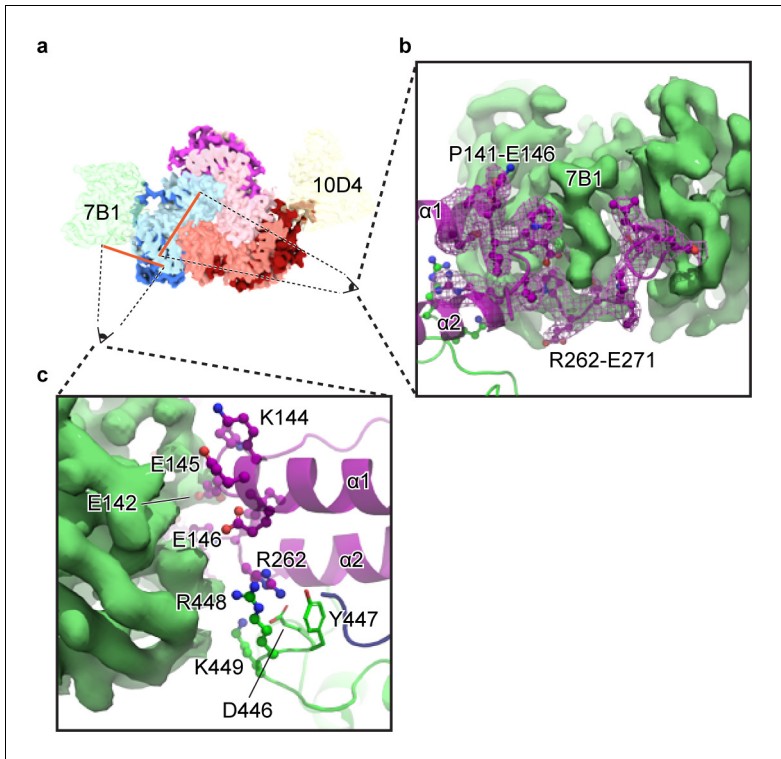

**Figure 6.** The 7B1 Fab binds to the α subunit making contacts with residues in the finger and thumb domains. (a) Top-down view of the ENaC_FL in complex with 7B1 and 10D4. The subunits and Fabs are colored as in *Figure 1*. (b) View of the interaction between the 7B1 Fab and the α2/α3 helices of the finger domain. The finger domain weaves within the binding region of the Fab. (c) Another view of the interaction between 7B1 and α subunit mediated by the finger and thumb domains. Acidic and basic residues that belong to the α subunit primarily mediate the interactions.

## Discussion

Here we employed single-particle cryo-EM to identify the structural determinants of subunit stoichiometry and arrangement in ENaC, and to illuminate the structural basis of ENaC modulation by $Na^+$ and proteolysis. Functional analysis of different combinations of ENaC subunits demonstrated that robust $Na^+$ currents were measured only when α, β, and γ were expressed together (*Canessa et al., 1994*). The first structure of human ENaC, ΔENaC, provided the first direct observation of the preferred assembly of the channel – counterclockwise α-β-γ when viewed from outside the cell. Our new structure, ENaC_FL, confirms the observed assembly and, with a more reliable placement of side chains, it deepens our understanding of the molecular principles that govern the heteromeric assembly of ENaC. The precise nature of how the subunits are positioned around the pseudo-three-fold axis involves an asymmetric arrangement of domains and unique molecular properties at interfaces.

The ECD of ENaC_FL deviates significantly from C3 symmetry. The swapped conformations of the β11-β12 linker in the β subunit and the β6-β7 aspartate and phenylalanine side chains in the α subunit are clear in the ENaC_FL map. The functional consequences of the swapped conformation of the β11-β12 linker in the β subunit are currently unknown but the findings provide a new direction for investigating the role of the β subunit in channel function. Furthermore, the presence of a phenylalanine adopting a different conformation in the α subunit from the tyrosines in the β and γ subunits, near the cation site confers a specialized function for the α subunit as the primary $Na^+$ sensor.

Additionally, the high-resolution information of the GRIP domains in ENaC_FL allowed us to investigate the specific interactions between the inhibitory peptides and their binding pockets in the channel. We note that the loss of strong features in the P3 strands in the cleaved state is in agreement with functional studies of the α subunit by Kashlan and colleagues (*Kashlan et al., 2010*). While equivalent studies in the γ subunit have not been performed, our focused-classification maps of

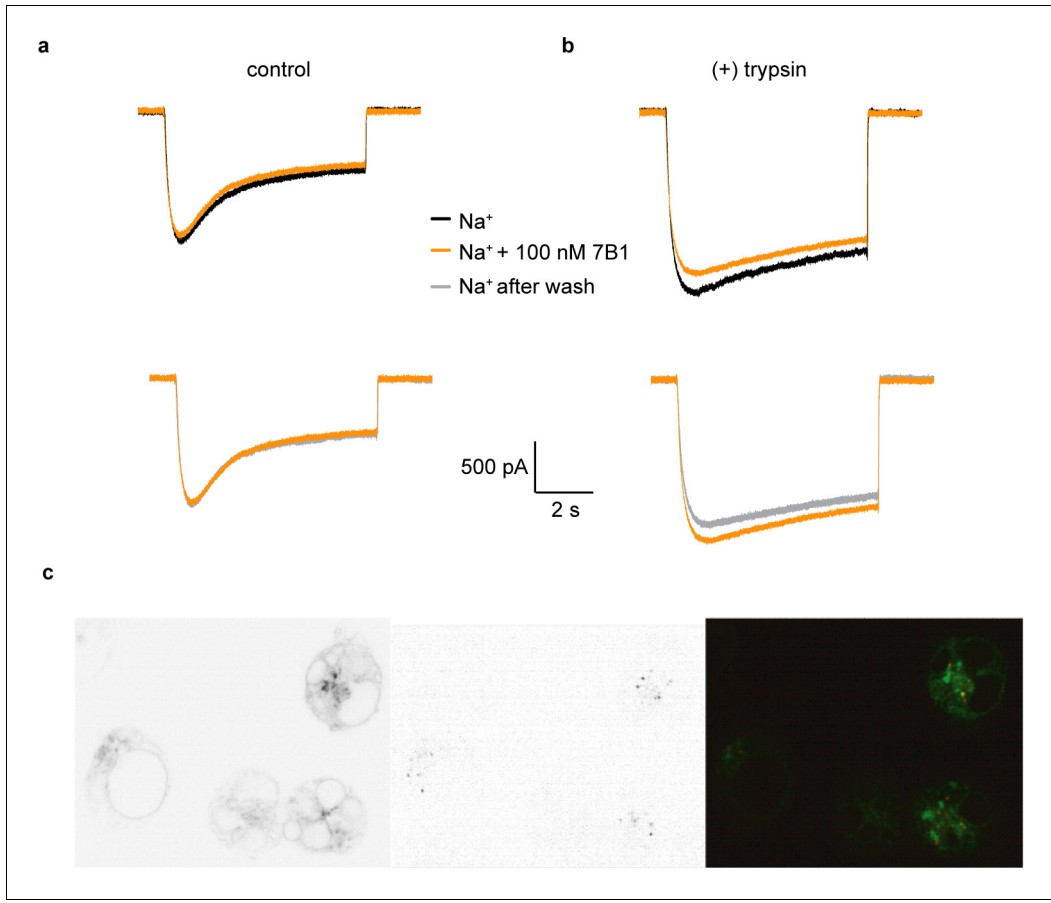

**Figure 7.** 7B1 binds to the α subunit independent of the cleavage state of the α subunit. (**a**) Whole-cell patch clamp measurements of ENaC-mediated Na$^+$ current indicate the 7B1 Fab does not alter current amplitude and shape. (**b**) Similar to the control current, 7B1 does not mediate acute effects of trypsin-cleaved ENaC. (**c**) Live confocal microscopy of HEK293S GNTI$^-$ cells expressing ENaC$_{FL}$ with eGFP fusion (left panel) are stained with TRITC labeled 7B1mAb (middle), recognizing the α subunit. The overlay of the GFP and TRITC channels (yellow, right panel) show that 7B1 mAb binds to ENaC$_{FL}$ that are expressed on the cell surface. Images were acquired at a pixel size of 0.13 μm for two different wavelengths at 488 nm and 561 nm. The samples were binned 2 × 2 and the exposure time was 400 ms for 488 nm and 1 s for 561 nm.

The online version of this article includes the following figure supplement(s) for figure 7:

**Figure supplement 1.** 7B1 binds to both uncleaved and cleaved ENaC$_{FL}$.

γGRIP indicate that similar structures and mechanisms exist between both the α and γ subunits. We speculate that the P3 strand anchors the N-terminal side of the α2 helix in place in the presence of the P1 peptide. After removal of P1, P3 is released, which destabilizes the α2 helix and β6/β7 interactions, disrupting the pocket for the novel α subunit Na$^+$-binding site we identified in the present study. This provides a structural explanation for the observed loss of Na$^+$ self-inhibition after proteolytic cleavage (*Sheng et al., 2006*). Surprisingly, we did not observe large-scale conformational differences in the α subunit between cleaved and uncleaved ECD maps. This is in contrast with the closed and open states of ASIC, in which the finger and thumb collapse in the open state. We confirmed that our tightly-binding 7B1 Fab does not trap the channel in a closed-like state. It is possible that the lack of observed conformational changes is a result of particle selection for an overall, closed structure, and then isolating those particles with a cleaved inhibitory peptide. It is possible that ENaC has a different gating mechanism from ASIC. Our maps highlight the importance of the α2 helix in mediating ENaC activity.

In this present study, we were unable to capture a fully cleaved state of γGRIP, which has a dominant effect on ENaC $P_O$ (*Carattino et al., 2008b*). The different classes derived from the focused

classification analysis of the γGRIP all demonstrate that unlike the αP1 peptide, the γP1 peptide spans the finger domain forging extensive contacts with the thumb (*Balchak et al., 2018*). Whether complete removal of the γP1 peptide gives rise to conformational changes at the finger/thumb interface will be addressed by resolving a structure of a fully cleaved γGRIP. It is of vital importance to resolve the TMD and CD, especially the pore-forming region of ENaC. A recent structure of ASIC in a lipid bilayer reveals that a highly conserved 'His-Gly' (HG) motif forms a reentrant loop that lines the lower ion permeation pathway (*Yoder and Gouaux, 2020*). The HG motif is found in all ENaC/Degenerin family members and is critical for ENaC function (*Gründer et al., 1997*). With better methods for isolating ENaC with stable TMD and CD in addition to further improvement of sample preparation, we hope to resolve the full channel to understand the mechanistic link between removal of inhibitory peptides in the ECD and channel gating.

# Materials and methods

**Key resources table**

| Reagent type (species) or resource | Designation | Source or reference | Identifiers | Additional information |
|---|---|---|---|---|
| Gene (*Homo sapiens*) | Amiloride-sensitive sodium channel subunit alpha isoform 1 | Synthetic | NCBI Reference Sequence: NP_001029.1 | Gene synthesized by BioBasic |
| Gene (*Homo sapiens*) | Amiloride-sensitive sodium channel subunit beta | Synthetic | NCBI Reference Sequence: NP_000327.2 | Gene synthesized by BioBasic |
| Gene (*Homo sapiens*) | Amiloride-sensitive sodium channel subunit gamma | Synthetic | NCBI Reference Sequence: NP_001030.2 | Gene synthesized by BioBasic |
| Cell line (*Homo sapiens*) | HEK293T/17 | ATCC | Cat #ATCC CRL-11268 | |
| Cell line (*Homo-sapiens*) | HEK293S GnTI- | ATCC | Cat #ATCC CRL-3022 | |
| Antibody | 7B1 mouse monoclonal | OHSU VGTI, Monoclonal Antibody Core | AB_2744525 | Isotype IgG2a, kappa, 1:2 molar ratio |
| Antibody | 10D4 mouse monoclonal | OHSU VGTI, Monoclonal Antibody Core | AB_2744526 | Isotype IgG1, kappa. 1:2 molar ratio |
| Recombinant DNA reagent | pEG BacMam | Gift from Eric Gouaux | Doi: 10.1038/nprot.2014.173 | |
| Chemical compound, drug | Amiloride hydrochloride hydrate | Sigma | Cat#: A7410 | |
| Chemical compound, drug | Phenamil Mesylate | Tocris | Cat#: 3379 | |
| Chemical compound, drug | Benzamil hydrochloride hydrate | Sigma | Cat#: B2417 | |
| Other | TRITC | ThermoFischer | Cat#: 46112 | |
| Software algorithm | HOTSPUR | Doi: 10.1017/s1431927619006792 | | |
| Software algorithm | MotionCor2 | Doi:10.1038/nmeth.4193 | SCR_016499 | https://emcore.ucsf.edu/ucsf-motioncor2 |
| Software algorithm | Ctffind4 | Doi: 10.1016/j.jsb.2015.08.008 | RRID:SCR_016732 | https://grigoriefflab.umassmed.edu/ctffind4 |
| Software algorithm | CryoSPARC | Doi:10.1038/nmeth.4169 | SCR_016501 | https://cryosparc.com/ |
| Software algorithm | cisTEM1.0.0 | Doi: 10.7554/eLife.35383 | SCR_016502 | https://cistem.org/ |
| Software algorithm | pyem | Doi: 10.5281/zenodo.3576633 | | https://github.com/asarnow/pyem |

*Continued on next page*

*Continued*

| Reagent type (species) or resource | Designation | Source or reference | Identifiers | Additional information |
|---|---|---|---|---|
| Software algorithm | Pymol | Pymol Molecular Graphics System, Schrodinger, LLC | RRID:SCR_000305 | http://www.pymol.org/ |
| Software algorithm | UCSF Chimera | Doi: 10.1002/jcc.20084 | RRID:SCR_004097 | http://plato.cgl.ucsf.edu/chimera/ |
| Software, algorithm | UCSF ChimeraX | Doi: 10.1002/pro.3235 | RRID:SCR_015872 | https://www.cgl.ucsf.edu/chimerax/ |
| Software, algorithm | Coot | Doi: 10.1107/S0907444910007493 | RRID:SCR_014222 | https://www2.mrc-lmb.cam.ac.uk/personal/pemsley/coot/ |
| Software, algorithm | Phenix | Doi:10.1107/S2059798318006551 | RRID:SCR_014224 | https://www.phenix-online.org/ |
| Software, algorithm | MolProbity | Doi:10.1107/S0907444909042073 | RRID:SCR_014226 | http://molprobity.biochem.duke.edu |

## Construct design

Two sets of constructs were designed for functional and structural studies. First, wild-type human α, β, and γ subunits were N-terminally fused with 8xHis tag, eGFP, and a thrombin recognition site (LVPRG); together, we refer to this set of constructs as ENaC$_{eGFP}$. The ENaC$_{eGFP}$ complex was ideal for whole-cell patch-clamp electrophysiology because the three eGFP per ENaC molecule facilitate in identifying ENaC-expressing cells. Second, for the biochemical aspects of the investigation, we put together another set of ENaC constructs in which the wild-type α and β subunits are untagged. As in ENaC$_{eGFP}$, the wild-type γ subunit is N-terminally fused with an 8xHis tag, eGFP, and a thrombin site, and together with WTα and WTβ make a heteromeric ENaC$_{FL}$. Because ENaC$_{FL}$ only contains one eGFP per ENaC molecule, we reduced eGFP contamination during the purification step when using eGFP nanobody for affinity purification.

## Generation and isolation of Fabs

The protocol for generation and isolation of Fabs are as described in *Noreng et al., 2018*. Mouse monoclonal antibodies 7B1 and 10D4 were generated using standard procedure by Dan Cawley at the Vaccine and Gene Therapy Institute (OHSU). The 7B1 and 10D4 mAbs were previously selected because they recognize tertiary epitopes of ENaC. The mAbs were purified, and their Fabs were generated by papain cleavage. Fab 7B1 was isolated by anion exchange using HiTrap Q HP column while Fab 10D4 was eluted using Protein A column to remove Fc.

## Expression and purification of ENaC-Fab complexes

Human embryonic kidney cells (HEK293T/17) were grown in suspension at a density of $2 - 4 \times 10^6$ cells/mL in Freestyle medium with 2% FBS and transduced with ENaC subunit virus to generate complexes ENaC$_{eGFP}$ and ENaC$_{FL}$ at a multiplicity of infection (MOI) of 1 and incubated at 37°C. 5 - 8 hr post transduction, amiloride was added to a final concentration of 1 µM, and cells were incubated at 30°C. After 24 – 48 hr, the cells were collected by centrifugation at 4790 xg for 20 min. The pellet was washed with 20 mM Tris, 200 mM NaCl and followed by a second round of centrifugation at 4790 xg for 15 min.

There were two approaches to purification of ENaC$_{FL}$ that the cryo-EM data set arrived from. In both purifications, GFP-cleaved ENaC$_{FL}$-diFab at pH 7.4 was the final purified complex used for cryo-EM sample preparation and will be referred to as ENaC$_{FL}$. In one purification, membranes were prepared and ENaC$_{FL}$ was purified from the prepared membranes, while in the second purification ENaC$_{FL}$ was purified from the cell pellet.

In the first purification, cells expressing ENaC$_{FL}$ were homogenized with a dounce homogenizer and sonicated in 20 mM Tris pH 7.4, 200 mM NaCl, 5 mM MgCl$_2$, 25 U/mL nuclease and protease inhibitors. Lysed cells were centrifuged at 9715 xg for 20 min and the resulting supernatant containing the membrane fractions were further centrifuged at 100,000 xg for 1 hr. Membrane pellets were resuspended and solubilized in 20 mM Tris pH 7.4, 200 mM NaCl, 2 mM ATP, 2 mM MgCl$_2$, 1%

digitonin (high purity, Millipore Sigma), 25 U/mL nuclease and protease inhibitors for 1 hr at 4°C. The solubilized fraction was isolated by ultracentrifugation 100,000 xg for 1 hr at 4°C.

In the second purification, cells expressing ENaC$_{FL}$ were homogenized with a dounce homogenizer in 20 mM HEPES pH 7.4, 150 mM NaCl, 2 mM MgCl$_2$, 25 U/mL nuclease and protease inhibitors. Homogenized cells were solubilized by adding the same buffer containing 2% digitonin (high purity, Millipore Sigma) and 4 mM ATP at 1 x initial volume (final volume 2x) for 2 hr at 4°C. The solubilized fraction was isolated by ultracentrifugation 100,000 xg for 1 hr at 4°C and supernatant was filtered through 0.45 μm filters.

Solubilized ENaC$_{FL}$ (from both purifications) was bound to GFP nanobody resin by batch binding for 2 hr at 4°C. ENaC$_{FL}$ bound to GFP nanobody resin was packed into an XK-16 column, and the column was washed with 20 mM Tris pH 7.4, 200 mM NaCl, 0.1% digitonin and 25 U/mL nuclease (second purification: 20 mM HEPES pH 7.4, 150 mM NaCl, 0.1% digitonin and 25 U/mL nuclease) followed by an additional wash of the same buffer containing 2 mM ATP. For elution, thrombin at 30 μg/mL and 5 mM CaCl$_2$ in the same buffer was applied to the column and incubated for 30 min. GFP-cleaved ENaC$_{FL}$ was eluted off with the same wash buffer and the eluted fractions were concentrated and then incubated with the Fabs 7B1 and 10D4 (DiFab complex) in a 1:2 molar ratio of ENaC$_{FL}$:Fab for 10 min, and clarified by ultracentrifugation 100,000 xg for 1 hr at 4°C. The supernatant was injected onto a Superose 6 Increase 10/300 GL column equilibrated in 20 mM Tris pH 7.4, 200 mM NaCl, 0.1% digitonin (second purification: 20 mM HEPES pH 7.4, 150 mM NaCl, 0.1% digitonin) to isolate the protein complex by size-exclusion chromatography. Monodispersed peaks were pooled and concentrated to 2 – 3 mg/mL.

## Image acquisition and data processing

Purified GFP-cleaved ENaC$_{FL}$-DiFab complexes at a concentration of 2 – 3 mg/mL was applied on holey-carbon cryo-EM grids which were glow discharged at 15mA for 60 s (Quantifoil Au 1.2/1.3 μm 300 mesh) prior to use. Grids were prepared using a Vitrobot Mark III (FEI) at 100% humidity and 12°C, where 3.5 μL of purified ENaC$_{FL}$-DiFab complexes were applied followed by a manual blot on the side of the grid. Then another 3.5 μL of purified ENaC$_{FL}$-DiFab complexes were applied before a wait time of 10 s, 3.5 s blot time at blot force 1, and then plunge frozen in liquid ethane cooled by liquid nitrogen.

Three large data sets were collected on the same microscope, a Titan Krios at the Multiscale Microscopy Core at OHSU, equipped with a Gatan K3 detector. One of the data sets, with a total of 9435 movies, were collected from the purification of ENaC$_{FL}$ solubilized from membranes, while the other two data sets were collected from ENaC$_{FL}$ purified directly from cells (see section 'expression and purification' for more details), one containing 9605 movies and the other containing 6153 movies. For all three data sets, movies were collected in super resolution mode with a pixel size of 0.415 Å. Total acquisition time was 3 s, and all three data sets were dose-fractionated to 60 frames with a dose rate of 1 e$^-$/Å$^2$/frame and total dose of 60 e$^-$/Å$^2$. Multishot with image shift between four holes was performed to speed up data collection using the automated acquisition program SerialEM (*Mastronarde, 2003*). Hotspur was used to initiate image alignment and ctf estimation during data collection (*Elferich et al., 2019*).

All data sets were binned 2 × 2 and motion corrected using motioncor2 (*Zheng et al., 2017*) with patch of 5 × 5. Each data set was processed individually using the software cryoSPARC v2 (*Punjani et al., 2017*) to determine the overall quality of final cryo-EM map before all three data sets were combined for final data processing. Defocus values were estimated using CTFFIND4 (*Rohou and Grigorieff, 2015*), and cryoSPARC blob picker was used for automated particle picking, initially resulting in 1,787,887 particles. Multiple rounds of 2D classification in cryoSPARC were performed where positive 2D classes of ENaC$_{FL}$ were saved, and particles belonging to false-positive classes were combined and re-classified by 2D classification to further reveal and include true ENaC$_{FL}$-diFab classes. After multiple rounds of 2D classification, a set of 453,875 particles was classified by cryoSPARCv2 *ab initio* and three rounds of 3D classification by heterogeneous refinement in cryoSPARCv2. To include as many true positive ENaC$_{FL}$-diFab particles as possible, 3D classes of false – positive particles went through additional 2D classification and positive ENaC$_{FL}$-diFab 2D classes were re-added for heterogeneous 3D classification.

The final data set containing 252,071 particles was processed using non-uniform refinement in cryoSPARC with default settings and C1 symmetry for a final 3D reconstruction with a Gold standard

Fourier Shell Correlation (GS FSC) resolution of 3.06 Å. In addition, the same particles were exported from cryoSPARC by using the pyem conversion script (csparc2star.py) (*Asarnow et al., 2019*), and then imported to cisTEM 1.0.0 (*Grant et al., 2018*). In cisTEM particles were sorted by 2D classification, and 248,079 particles were refined with a mask that contained the ECD only to a solvent adjusted FSC of 3.11 Å. The final 3D map from 252,071 particles created in cryoSPARC v2 was used for model building and refinement. The improved resolution is potentially due to the advancement of the detector that was used for data collection (Gatan K2 switched to a Gatan K3 detector), as well as improvement of sample preparation where ENaC$_{FL}$ grids were imaged in regions containing thinner ice.

To separate cleaved states of ENaC$_{FL}$, focused classification (only refining the translational x and y parameters in cisTEM 1.0.0) was performed in the GRIP domain of the α and γ subunits. Subsequent classes obtained from focused classification were imported to cryoSPARC and *ab initio* was performed followed by non-uniform heterogeneous refinement to confirm the missing densities.

## Model building

The extracellular coordinates of the ΔENaC structure and the antigen-binding domains of 7B1 and 10D4 (PDB code: 6BQN [*Noreng et al., 2018*]) were docked into the cryo-EM map using Chimera (*Pettersen et al., 2004*). The coordinates were then manually inspected and adjusted using the computer program COOT (*Emsley and Cowtan, 2004*). The overall improved map quality shows many well-defined features that were not resolved in the ΔENaC map. These features include additional residues in the α- and γ-P1 peptides, Na$^+$ ion, and N-acetylglucosamines (GlcNac). The final model contains all residues proposed to comprise the inhibitory peptides, LPHPLQRL and RFSHRIPLLIF, in the α- and γ-GRIP domains, respectively (*Carattino et al., 2008a*; *Passero et al., 2010*). Furthermore, seven glycosylation sites were modeled: two in α, four in β, and one in γ subunit.

Due to the lack of map features corresponding to the segments that connect the GRIP domains to the α1 and α2 helices, the loops were not included in the final model. Importantly, the ENaC$_{FL}$ TMD was also excluded from the final model. While the 2D class averages and 3D maps demonstrate micelles features, which suggest the presence of the ENaC$_{FL}$ TMD, the ion channel portion of the complex was not resolved. Overall, the following residues were modeled into the ENaC$_{FL}$ cryo-EM map: residues 114–166, 183–191, 223–541 in α, 78–131, 139–167, 179–481, 486–512 in β, and 80–133, 152–164, 200–521 in γ. Iterative rounds of manual building and real-space refinement were conducted using COOT and PHENIX (*Adams et al., 2011*), respectively. The final model was determined to have good stereochemistry as assessed by MolProbity (*Chen et al., 2010*). Distance measurements and figures were made using the software Pymol (*Schrodinger, LLC, 2015*) and chimeraX (*Goddard et al., 2018*).

## Confocal fluorescence microscopy

Confocal fluorescence microscopy was performed as previously reported (*Noreng et al., 2018*). The antibody was conjugated to TRITC at a final dye:protein molar ratio of 3.7:1 in TBS.

## Western blotting

For western blots, ENaC$_{FL}$ was purified as described above (solubilized with 20 mM DDM and 3 mM CHS instead of 1% digitonin). For the biochemical characterization of ENaC$_{FL}$ as shown in *Figure 1— figure supplement 1*, the following polyclonal antibodies were used: sc-21012 (αENaC), ABclonal A1765 (βENaC), and ABclonal A15097 (γENaC). To validate purified ENaC$_{FL}$ samples treated with trypsin, we also used western blotting as shown in *Figure 7—figure supplement 1*. The sample was split into groups, one kept untreated while the other was treated with 25 μg/mL of trypsin for 5 min at room temperature. Both samples were injected individually onto a Superose 6 Increase 10/300 column. The peak fractions from each condition were collected, pooled, and split into two groups. The first group was concentrated and prepared for FSEC and western blotting. The second group was concentrated and diluted multiple times with 0.5 mM DDM, 75 μM CHS, 20 mM HEPES pH 7.4, and 150 mM KCl to attain a NaCl concentration of approximately 0.24 mM and a KCl concentration of 149.76 mM. As a result, there were four total samples: uncleaved ENaC$_{FL}$ in Na$^+$ or K$^+$ and cleaved ENaC$_{FL}$ in Na$^+$ or K$^+$. SDS-PAGE samples of 2.9–3.2 μg each (the same amount within a blot) were loaded into the wells of 4 – 15% Tris-HCl Criterion Precast Gel. Proteins were

electrophoresed at 180 V for 60 min and then blotted onto a nitrocellulose membrane at 80 V for 40 min. Membranes were blocked overnight at 4°C while rocking in 5% nonfat dry milk (NFDM). For staining, the primary antibody used was either αENaC (6 µg/ blot, rabbit polyclonal Ab to SCNN1A raised against amino acids 131–225, sc-21012) or γENaC (11 µg/blot, rabbit polyclonal Ab to SCNN1G raised against amino acids 100–200, ab133430). Primary antibodies were left on the membrane for 2 hr at room temperature while rocking. IRDye 680RD Goat anti-mouse IgG (LI-COR, 925–68070) was used as the secondary antibody on both blots. The secondary antibody was diluted to 1:25000 (1 µg/ 25 mL TBST) and allowed to bind for 1 hr at room temperature while rocking. The blots were imaged on an Odyssey western blot detection system.

## Whole cell patch clamp experiments

HEK293T/17 cells were grown in suspension at a density of $2 - 4 \times 10^6$ cells/mL in Freestyle medium with 2% FBS and transduced with the virus (ENaC$_{eGFP}$) at a multiplicity of infection (MOI) of 1 and incubated at 37°C. After approximately 5 hr the transduced suspension cells were incubated in the presence of 500 nM phenamil mesylate at 30°C for 12 – 14 hr. About 2–3 hr before recording, cells were transferred to wells containing glass coverslips at a density $0.3 - 0.5 \times 10^6$ cells/mL and in Dulbecco's Modified Eagle Medium supplemented with 2% FBS and 500 nM phenamil mesylate. Whole cells recordings were carried out 17 – 24 hr after transduction. Pipettes were pulled and polished to $2.5 - 3.5$ MΩ resistance and filled with internal solution containing (in mM): 150 KCl, 2 MgCl$_2$, 5 EGTA, and 10 HEPES pH 7.4. For IC$_{50}$ experiments, external solutions that were used contained (in mM): 150 NMDGCl or NaCl, 2 MgCl$_2$ and CaCl$_2$, and 10 HEPES pH 7.4. Increasing concentrations (1 nM, 10 nM, 100 nM, 1 µM, 10 µM, 100 µM) of amiloride, phenamil mesylate, or benzamil were added to the solution containing 150 mM NaCl. The macroscopic ENaC current was determined as the blocker-sensitive Na$^+$-current that was blocked by 100 µM amiloride, phenamil mesylate, or benzamil. To determine the voltage sensitivity of each blocker, steps of +20 mV, from a starting holding potential at −60 mV up to 0 mV was performed for each experiment and the IC$_{50}$ was determined for each voltage step (−60 mV, −40 mV, −20 mV and 0 mV). All recording experiments were repeated independently five times.

For whole cell patch clamp experiments to determine the effect of the monoclonal antibody (mAb) 7B1, current amplitudes were measured before and after addition of 100 nM of 7B1 for 3 min. Cells were then washed with buffer before application of trypsin (5 µg/mL) for 5 min to increase amiloride-sensitive Na$^+$ currents. Post treatment with trypsin, cells were incubated with mAb 7B1 at a final concentration of 100 nM for 3 min. The amiloride-sensitive Na$^+$-current was recorded before and after incubation with mAb.

## Acknowledgements

We thank L Vaskalis for help with figure design. Cryo-EM data were collected at the Pacific Northwest Center for Cryo-EM (PNCC) and Multiscale Microscopy Core (MMC) at OHSU. Confocal data were collected at the Advanced Light Microscopy Core at OHSU. This work was supported by the National Institute of Health and the American Heart Association (AHA) (IB, DP5OD017871 and 19TPA34760754) and the AHA (SN, 18PRE33990205) and the National Science Foundation (NSF) (AH, DGE-1937961).

## Additional information

### Funding

| Funder | Grant reference number | Author |
| --- | --- | --- |
| National Institutes of Health | DP5OD017871 | Isabelle Baconguis |
| American Heart Association | 19TPA34760754 | Isabelle Baconguis |
| American Heart Association | 18PRE33990205 | Sigrid Noreng |
| National Science Foundation | DGE-1937961 | Alexandra Houser |

The funders had no role in study design, data collection and interpretation, or the decision to submit the work for publication.

## Author contributions

Sigrid Noreng, Conceptualization, Data curation, Formal analysis, Funding acquisition, Validation, Investigation, Visualization, Methodology, Writing - original draft, Writing - review and editing; Richard Posert, Data curation, Formal analysis, Validation, Investigation, Visualization, Writing - original draft, Writing - review and editing; Arpita Bharadwaj, Conceptualization, Data curation, Formal analysis, Validation, Visualization, Writing - review and editing; Alexandra Houser, Data curation, Formal analysis, Validation, Visualization, Writing - review and editing; Isabelle Baconguis, Conceptualization, Resources, Data curation, Formal analysis, Supervision, Funding acquisition, Validation, Investigation, Visualization, Methodology, Writing - original draft, Project administration, Writing - review and editing

## Author ORCIDs

Sigrid Noreng https://orcid.org/0000-0001-5767-1399
Richard Posert https://orcid.org/0000-0001-9010-2104
Arpita Bharadwaj https://orcid.org/0000-0002-3867-7610
Alexandra Houser https://orcid.org/0000-0001-5516-6225
Isabelle Baconguis https://orcid.org/0000-0002-5440-2289

## Decision letter and Author response

Decision letter https://doi.org/10.7554/eLife.59038.sa1
Author response https://doi.org/10.7554/eLife.59038.sa2

## Additional files

### Supplementary files

• Transparent reporting form

### Data availability

All cryo-EM maps have been deposited in the Electron Microscopy Data Bank under the accession code EMD-21896 for ENaC. Model coordinates have been deposited in the Protein Data Bank under the accession code 6WTH.

The following datasets were generated:

| Author(s) | Year | Dataset title | Dataset URL | Database and Identifier |
|---|---|---|---|---|
| Baconguis I | 2020 | cryo-EM maps for ENac | http://www.ebi.ac.uk/pdbe/entry/emdb/EMD-21896 | Electron Microscopy Data Bank, EMD-21896 |
| Baconguis I | 2020 | Model coordinates | http://www.rcsb.org/structure/6WTH | RCSB Protein Data Bank, 6WTH |

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
