## [Decision Letter]

**Acceptance summary:**

This is a follow-up to a paper published previously from the same group reporting an atomic model for the ENaC channel using cryo-EM methods. In this work, the authors report results from a full-length channel construct, as opposed to the truncated subunits previously used. The new structure has increased resolution in the extracellular aspects of the protein, provides an explanation for the heterotrimeric organization and overall counter clockwise arrangement of the subunits. These are significant advances that should be of interest to the field.

**Decision letter after peer review:**

Thank you for submitting your article "Molecular principles of assembly, activation, and inhibition in epithelial sodium channel" for consideration by *eLife*. Your article has been reviewed by three peer reviewers, and the evaluation has been overseen by Sriram Subramaniam as Reviewing Editor and Richard Aldrich as the Senior Editor. The following individuals involved in review of your submission have agreed to reveal their identity: Lawrence Palmer (Reviewer #1); Erhu Cao (Reviewer #3).

The reviewers have discussed the reviews with one another and the Reviewing Editor has drafted this decision to help you prepare a revised submission.

Since each of the reviewers emphasized distinct points in their comments, we are enclosing them individually.

Reviewer #1:

This is a follow-up to a paper published in 2018 that reported the first atomic-level structure of ENaC. The authors report results from a full-length channel construct, as opposed to the truncated subunits previously used. Furthermore the new construct can be proteolytically cleaved, permitting the study of an important regulatory process. The new structure has increased resolution in the extracellular aspects of the protein, allowing conclusions to be drawn regarding subunit assembly. The work explains why the channel is a heterotrimer, and why the preferred arrangement of subunits in a counter clockwise direction. A possible site of interaction of extracellular Na with the channel is identified. These are significant advances that should be of interest to the field.

I have a number of suggestions for revision.

1) The new structure includes the cytoplasmic domains of the channel subunit, but nothing is said about them. Presumably they are not sufficiently resolved but this should be stated explicitly.

2) The Na binding site is a major part of the story. The density map showing this site should be shown as a part of Figure 3, to convince the reader that the density ascribed to Na is real. Furthermore, the position of this site within the whole channel structure should be illustrated.

3) The identification of this site with the Na self-inhibition process cannot be made definitively at this point. Although the authors question the ion selectivity of this site, the original description of the phenomenon (Fuchs, Larsen and Lindemann, 1977) defines it as the response to replacing K with Na, implying strong selectivity between these ions. If this is the true site at which Na inhibits its own conductance, one would expect either a change in ion binding or a change in structure when K replaces Na in the medium. If this experiment cannot be done I suggest that the conclusions be softened.

4) In Figure 5, the conclusions rest on a structure from a small fraction (~10%) of channels. This is surprising since the Western blot in Figure 1—figure supplement 4D seems to show that most of the α subunits are in the cleaved state. I also wonder about the identification of the bands in this blot. To what part of the subunit does the antibody bind? How can it recognize both cleavage fragments? Please state the sources of the antibodies.

5) Figure 8 was not helpful to me in understanding the proposed mechanism. It should be improved or dropped.

Reviewer #2:

This manuscript by Noreng and colleague investigates the assembly and activation/inhibition mechanism of the human epithelial sodium channel (ENaC). The channel is activated via proteolysis on their extracellular domains. Using cryo-EM, the authors obtained maps of the entire protein as well as partially cleaved channels. Based on these, they were able to proposed a model of the hetero-trimeric assembly of the channel, a sodium binding site and mechanism of inhibition by peptides. These results are significant contributions to the ENaC field. While the experiments were done carefully, the presentation of the results in the manuscript is somewhat difficult to follow. Significant clarification and modification are required before its publication in *eLife*.

1) The nomenclatures of the structural elements in subunits are confusing. While the three protein subunits are already named α, β and γ, the structural elements in all the subunits are again called α1 and β6, etc. αGly225 is obviously in the α-subunit, but α1, α2, β6 and β7 appear in many places in the text without defining which subunit they belong to, making the text difficult to follow.

2) The number of possible conformations after partial proteolysis is unclear. The three protein subunits can be cleaved, twice, once and once, respectively. Therefore, the possible number of their states are 3 (α: 0, 1, 2 cuts), 2 (β: 0, 1) and 2 (γ: 0, 1), respectively. The total number of their possible combinations is 12 even if the order of cleavage at the two α sites are not considered. Explanations are needed to show why only 4 conformations are considered (subsection “Investigation of the GRIP domains”, second paragraph).

3) The data presentation for cryo-EM data collection and processing in Table 1 in inadequate. Instead of software used, more details on data processing and model refinement should be included.

Reviewer #3:

Noreng et al. report a 3 Å resolution structure of the ENaC channel. With greatly improved resolution than the previous structure determined by the same group, the authors can now define the molecular mechanisms that underlie the assembly of the triheteromeric ENaC channel. Moreover, they were also able to visualize an inhibitory Na^+^ binding site within the α subunit that is known to play crucial roles in the regulation of ENaC activity. The structure also revealed the GRIP domain that confers channel activation upon cleavage by proteases. Overall, although structure of the transmembrane domain that harbors the ion conduction pathway is yet to be determined, these findings represent an important step toward understanding the structure-function relationship of the ENaC channel.

---

## [Author Response]

Reviewer #1:[…] I have a number of suggestions for revision.1) The new structure includes the cytoplasmic domains of the channel subunit, but nothing is said about them. Presumably they are not sufficiently resolved but this should be stated explicitly.

We appreciate this point and have revised the following sentences in our text to include the cytosolic domains (CD), it now reads:

“However, the transmembrane domain (TMD) and the cytosolic domain (CD) were not resolved;…”

“Therefore, we did not include the TMD and CD portions in the ENaC_FL_ structure.”

We also emphasized the importance of visualizing the CD in the Discussion section and it now reads, “It is of vital importance to resolve the TMD and CD, especially the pore-forming region of ENaC.”

Additionally we have included two sentences briefly mentioning the most recent structure of ASIC: “A recent structure of ASIC in a lipid bilayer reveal that a highly conserved ‘His-Gly’ (HG) motif forms a reentrant loop that lines the lower ion permeation pathway. The HG motif is found in all ENaC/Degenerin family members and is critical for ENaC function. With better methods for isolating ENaC with stable TMD and CD in addition to further improvement of sample preparation, we hope to resolve the full channel to understand the mechanistic link between removal of inhibitory peptides in the ECD and channel gating.”

2) The Na binding site is a major part of the story. The density map showing this site should be shown as a part of Figure 3, to convince the reader that the density ascribed to Na is real. Furthermore, the position of this site within the whole channel structure should be illustrated.

We are grateful to the reviewer for pointing this out and have added a panel to show a view of the full channel viewed from the extracellular side. This panel is now Figure 3A. In this new panel, we have also added a label to show the putative cation site. Furthermore, we have modified the panel, now Figure 3B, showing the α subunit β6- β7 loop electrostatic potential. We have removed the sphere representation from the model to emphasize the presence of the map feature we observe in the 3D map of ENaC_FL_.

3) The identification of this site with the Na self-inhibition process cannot be made definitively at this point. Although the authors question the ion selectivity of this site, the original description of the phenomenon (Fuchs, Larsen and Lindemann, 1977) defines it as the response to replacing K with Na, implying strong selectivity between these ions. If this is the true site at which Na inhibits its own conductance, one would expect either a change in ion binding or a change in structure when K replaces Na in the medium. If this experiment cannot be done I suggest that the conclusions be softened.

We appreciate the reviewer pointing out that we cannot fully conclude the identification of the putative Na^+^ site as the true Na^+^ site without additional experiments supporting this. Obtaining the structure of ENaC where Na^+^ is replaced by K^+^ is important in order to determine whether there is a change in ion binding and/or structural change in absence of Na^+^. This experiment cannot be done within a reasonable time frame, but we are committed to future studies addressing these concerns. We have modified the text to read, “We speculate that this map feature is a cation, perhaps a Na^+^ ion, based on the surrounding residues, the above-mentioned functional studies, and the presence of high Na^+^ (150 mM) during purification (Figure 3B). […] Indeed, definitive identification of this feature as the Na^+^ self-inhibition site would require resolving the structure of ENaC in the presence of K^+^ and determining if there are any associated structural changes.”

4) In Figure 5, the conclusions rest on a structure from a small fraction (~10%) of channels. This is surprising since the Western blot in Figure 1—figure supplement 4D seems to show that most of the α subunits are in the cleaved state. I also wonder about the identification of the bands in this blot. To what part of the subunit does the antibody bind? How can it recognize both cleavage fragments? Please state the sources of the antibodies.

We appreciate the reviewer for raising this concern. The polyclonal antibodies used to detect the different ENaC subunit bands are as follows: αENaC (Santa Cruz Biotechnology, Inc, SC-21012), βENaC (ABclonal, A-1765), and γENaC (ABclonal A15097). Author response image 1 shows a schematic that demonstrates the epitope for the α subunit polyclonal antibody ENaC, SC-21012, based on the information provided by the vendor.

**Author response image 1. sa2fig1:** Schematic illustration of human ENaC-α subunit (hENaC-α). The two furin sites are illustrated as white dotted lines. The location of the epitope for the polyclonal antibody SC-210112 is illustrated in the schematic.

The sources of the antibodies are stated in the figure legend of Figure 1—figure supplement 1D, (α subunit), (β subunit), and (γ subunit). Additionally, we have updated the Materials and methods section to include the sources of all polyclonal antibodies that were used in each Western blot experiment, as well as updated the figure legend of Figure 7—figure supplement 1 where we also used polyclonal antibodies to detect α and γ subunits.

We have also added in the text to provide three possible reasons for the results of the Western blot and focused classification experiments. It reads, “While on the one hand the Western blot analysis showed a large population of cleaved α subunit, on the other hand, the focused classification analysis demonstrated a small population of the fully cleaved class. […] Thus, the population of particles used for SDS-PAGE and Western blot analyses is not the same as the population used for focused classification. Nevertheless, implementing focused classification resulted in 3D maps that demonstrate differences in map features in the GRIP domain.”

5) Figure 8 was not helpful to me in understanding the proposed mechanism. It should be improved or dropped.

We appreciate this comment. We have removed the figure.

Reviewer #2:[…] Significant clarification and modification are required before its publication in eLife.1) The nomenclatures of the structural elements in subunits are confusing. While the three protein subunits are already named α, β and γ, the structural elements in all the subunits are again called α1 and β6, etc. αGly225 is obviously in the α-subunit, but α1, α2, β6 and β7 appear in many places in the text without defining which subunit they belong to, making the text difficult to follow.

We agree that the nomenclature is inconsistent, leading to confusion when reading the text. The use of either ENaC domains or specific α helices and β strands within each ENaC subunit creates difficulties when reading the text. To improve clarity, we have defined the different domains in the text (Results). Furthermore, we have modified the text so when referring to interactions within subunit interfaces we either refer to the ENaC subunit and α helix (for example, γ-α1 helix refer to the α1 helix in the γ subunit), or refer to the specific residues directly (βVal474). Additionally, we have labeled the residues in Figure 1 to provide more clarity. We have also added the secondary structure terms “helix” or “strand” after “α” or “β,” respectively, to refer to the structural elements within ENaC subunits. All these changes have been added throughout the text.

2) The number of possible conformations after partial proteolysis is unclear. The three protein subunits can be cleaved, twice, once and once, respectively. Therefore, the possible number of their states are 3 (α: 0, 1, 2 cuts), 2 (β: 0, 1) and 2 (γ: 0, 1), respectively. The total number of their possible combinations is 12 even if the order of cleavage at the two α sites are not considered. Explanations are needed to show why only 4 conformations are considered (subsection “Investigation of the GRIP domains”, second paragraph).

We apologize for the confusion. We have modified the text to clearly explain the basis of our approach. First, we have added a sentence to clearly state that the β subunit does not have any protease sites. It now states, “Of note, the β subunit does not have canonical protease sites.” This information is once again mentioned, “The first structure of ENaC, referred to as ΔENaC which comprised of subunits with truncated amino and carboxy termini and other mutations in the ECD, demonstrated that all GRIP domains, including the protease-insensitive βGRIP, adopt similar anti-parallel β strand architecture (Noreng et al., 2018).”

In terms of conformational states, we would like to clarify that our goal for implementing focused classification is to identify the different cleavage states, not conformational states. We, thus, looked for presence or absence of the inhibitory peptide. Once cleavage states were identified, we proceeded to ask whether there were detectable conformational changes, which in this current study, we were unable to measure. We have added in the text, “Using focused classification, we aimed to identify the different cleavage states – absence or presence of the inhibitory peptide – that our SDS-PAGE analysis suggests to exist.”

3) The data presentation for cryo-EM data collection and processing in Table 1 in inadequate. Instead of software used, more details on data processing and model refinement should be included.

We recognize that the cryo-EM data collection and processing may appear inadequate. The table of the data processing and model refinement follows standard table configurations for cryo-EM data processing. However, we agree that information of model building contain less details. We have, thus, included additional lines in our table (initial model: 6BQN, Non-hydrogen atoms: 11,740, Ligands (Na, NAG): 1,10, Rotamer outliers: 0.84%) (see Table 2). We also included more details to explain the data processing strategy in the figure legends of Figure 1—figure supplements 2 and 3. The description provides more detailed information as to how we obtained the final cryo-EM map of ENaC_FL_. The revised figure legends are as follows:

“Figure 1—figure supplement 2. Cryo-EM initial data processing workflow. […] The sixth class was further classified by 2D classification to recover true-positive ENaC_FL_ classes, and added to a final stack of 315,477 particles.”

Figure 1—figure supplement 3. Cryo-EM data processing for the final map. […] This refined particle stack was also imported to cisTEM 1.0.0, classified by 2D classification, before final 3D refinement of 248,079 particles resulted in a cryo-EM map at a resolution of 3.11 Å (solvent-adjusted FSC).”